# The use of early warning system scores in prehospital and emergency department settings to predict clinical deterioration: A systematic review and meta-analysis

Gigi Guan[1,2]*, Crystal Man Ying Lee[3,4], Stephen Begg[5], Angela Crombie[6], George Mnatzaganian[1,7]

1 Rural Department of Community Health, La Trobe Rural Health School, La Trobe University, Bendigo, Victoria, Australia, 2 Department of Rural Health, Faculty of Medicine, Dentistry and Health Sciences, University of Melbourne, Shepparton, Australia, 3 School of Population Health, Curtin University, Perth, Western Australia, Australia, 4 School of Psychology and Public Health, La Trobe University, Melbourne, Victoria, Australia, 5 Violet Vines Marshman Centre for Rural Health Research, La Trobe University, Bendigo, Victoria, Australia, 6 Research & Innovation, Bendigo Health, Bendigo, Victoria, Australia, 7 The Peter Doherty Institute for Infection and Immunity, Melbourne, Victoria, Australia

* gigiaus@gmail.com

**Data Availability Statement:** All relevant data are within the paper and its Supporting Information files.

## Abstract

### Background

It is unclear which Early Warning System (EWS) score best predicts in-hospital deterioration of patients when applied in the Emergency Department (ED) or prehospital setting.

### Methods

This systematic review (SR) and meta-analysis assessed the predictive abilities of five commonly used EWS scores (National Early Warning Score (NEWS) and its updated version NEWS2, Modified Early Warning Score (MEWS), Rapid Acute Physiological Score (RAPS), and Cardiac Arrest Risk Triage (CART)). Outcomes of interest included admission to intensive care unit (ICU), and 3-to-30-day mortality following hospital admission. Using DerSimonian and Laird random-effects models, pooled estimates were calculated according to the EWS score cut-off points, outcomes, and study setting. Risk of bias was evaluated using the Newcastle-Ottawa scale. Meta-regressions investigated between-study heterogeneity. Funnel plots tested for publication bias. The SR is registered in PROSPERO (CRD42020191254).

### Results

Overall, 11,565 articles were identified, of which 20 were included. In the ED setting, MEWS, and NEWS at cut-off points of 3, 4, or 6 had similar pooled diagnostic odds ratios (DOR) to predict 30-day mortality, ranging from 4.05 (95% Confidence Interval (CI) 2.35–6.99) to 6.48 (95% CI 1.83–22.89), p = 0.757. MEWS at a cut-off point ≥3 had a similar DOR when predicting ICU admission (5.54 (95% CI 2.02–15.21)). MEWS ≥5 and NEWS ≥7 had DORs of 3.05 (95% CI 2.00–4.65) and 4.74 (95% CI 4.08–5.50), respectively, when

**Funding:** The author(s) received no specific funding for this work.

**Competing interests:** The authors have declared that no competing interests exist.

predicting 30-day mortality in patients presenting with sepsis in the ED. In the prehospital setting, the EWS scores significantly predicted 3-day mortality but failed to predict 30-day mortality.

## Conclusion

EWS scores' predictability of clinical deterioration is improved when the score is applied to patients treated in the hospital setting. However, the high thresholds used and the failure of the scores to predict 30-day mortality make them less suited for use in the prehospital setting.

## Introduction

Initially used in the intensive care unit (ICU), the Early Warning System (EWS) scores have been employed in multiple healthcare facilities, including hospital wards, the emergency department (ED), and prehospital community settings [1, 2]. These scores primarily aim to detect the clinical deterioration in patients by tracking their vital signs, with high EWS scores triggering a response to prevent any potential clinical decline. It has been observed that patients' vital signs usually change before any clinical deterioration [3, 4]. Consequently, if early and timely interventions are adequately performed, adverse outcomes of patients may be prevented. The earliest EWS score was validated in 1981 for only the ICU patients. Modifications over time were developed to suit various hospital inward settings, with some being more specific to certain conditions such as blunt trauma or sepsis [5–8]. However, the fundamentals of the scores have generally been consistent and involve measurements of variations in vital signs e.g., systolic blood pressure, oxygen saturation, temperature, heart rate, and Glasgow Coma Scale, which are used to calculate a cumulative score. Other versions of the EWS employ advanced therapies, laboratory testing, and demographic information of patients [2, 9]. The EWS scores constitute a standardised practice across prehospital and hospital settings in the UK [6] and in hospital settings in some parts of Australia.

Numerous systematic reviews and meta-analyses have identified optimal EWS scores in hospital settings (e.g., wards or the ED) [3, 10–14]. However, only a limited number of reviews focused on prehospital settings [15, 16]. Available systematic reviews demonstrate that EWS scores can be utilised to potentially improve patient outcomes [11–15] but, since several EWS scores are used across different settings, it is unknown which score should be used in the ED or prehospital setting to best predict clinical deterioration [17, 18]. Furthermore, the selection of EWS scores of best cut-off points to accurately predict outcomes such as short-term and long-term mortality or ICU admission has not been performed [6–8].

This systematic review and meta-analysis aimed to estimate the pooled odds of predicting clinical deterioration in hospitalised patients, including short (≤3-day) and long-term (≤30-day) mortality and ICU admission, by stratifying the EWS score cut-off points as used in the ED and prehospital settings. Length of stay in hospital and cardiac or respiratory arrests were also investigated.

## Methods

We reviewed the five most-used EWS scores in the ED or prehospital settings, including the National Early Warning Score (NEWS) and its updated version, the National Early Warning Score 2 (NEWS2), the Modified Early Warning Score (MEWS), the Rapid Acute Physiological

Score (RAPS) and the Cardiac Arrest Risk Triage (CART) [10, 12–15, 19]. These scores were selected since they rely solely on observations readily available to health care professionals in the prehospital setting and are easy to calculate and apply. Thus, we avoided the costly and time-consuming pathological and other complex testings, which are less frequently performed in the prehospital environment. A PICO framework was used to inform the literature search strategy (S1 File).

## Protocol and registration

The Preferred Reporting Items for Systematic Reviews and Meta-Analyses (PRISMA) statement was used to design and report this systematic review. The protocol was registered with the PROSPERO-International register of systematic reviews (registration number of CRD42020191254).

## Inclusion and exclusion criteria

Original research following experimental, quasi-experimental, or observational designs using EWS scores in either the ED or prehospital setting were eligible for inclusion. Additionally, studies were included if they reported on patients aged ≥14 years presenting with medical conditions including sepsis or injury associated conditions, with study outcomes including in-hospital mortality within 3–30 days of hospitalisation, cardiac arrest/respiratory arrest, and length of stay. Only studies published in English were included with no restriction to the year of publication.

Studies reporting on obstetric, maternal, or palliative care patients were excluded. Furthermore, articles focusing solely on Severe Acute Respiratory Syndrome Coronavirus 2 infections and rare conditions (e.g., portal hypertension) were excluded.

## Search strategy

Four databases (CINAHL, Embase, PubMed/MEDLINE, and Web of Science) were systematically searched in February 2021. Key search terms included prehospital/ambulance/paramedic, ED/emergency room, and the selected EWS scores. All synonyms and MeSH terms were included in the searches (S2 File). In addition, the references of the identified articles were manually searched for additional articles that might have been missed in the electronic searches.

## Selection process and data extraction

Three co-authors (GG, GM, CL) independently screened all articles based on title and abstract. In addition, author GG screened all potential articles cross-checked by CL (80%) and GM (20%). Any disagreements or conflicts were discussed to make the final decision for inclusion.

Data extracted included author name, year and country of publication, study setting (ED or prehospital), sample size, study outcomes, EWS scores, their cut-off points and sensitivity and specificity in predicting the desired study outcomes, mean or median age of study population, sex proportion, study design, and study inclusion and exclusion criteria. The Newcastle-Ottawa Scale was used to assess the methodological quality of the included articles [20]. GG and SB independently conducted the assessments, with conflicts resolved after discussion with co-authors.

## Statistical analysis

An odds ratio (OR) was computed as a summary measure of the predictive accuracy of each EWS from (TP x TN) / (FP x FN), in which TP, TN, FP, and FN respectively express true

positive, true negative, false positive and false negative. The confidence interval (CI) of the OR was estimated as:

$$CI = \exp\left(\log(OR) \pm Z\alpha/2 * \sqrt{1/TP + 1/TN + 1/FP + 1/FN}\right),$$

where $Z\alpha/2$ denotes the critical value of the normal distribution at $\alpha/2$ (e.g., for a CI 95%, $\alpha$ is 0.05, and the critical value reaches 1.96).

To express the diagnostic accuracy of the EWS scores, the log ORs, together with their corresponding log standard errors, were meta-analysed using DerSimonian and Laird random-effects models [21]. The analyses were conducted by the EWS scores cut-off points, patient outcomes (short- and long-term mortality from hospital admission and ICU admission) and study setting (ED or prehospital). We defined short-term mortality as death within three days of admission and long-term mortality as death within 30 days of admission. The heterogeneity between studies was estimated using $I^2$ statistic. Meta-regressions were constructed to quantify the proportion of variances between studies, as explained by sample size, sex proportion, and age. Deeks' funnel plot asymmetry test was used to test publication bias. Pooled diagnostic ORs were compared after converting them to z scores. The p values were estimated using the normal distribution table. Sensitivity analyses were conducted by risk of bias.

All analyses were conducted using Stata/SE 15.1 (Stata Corp LP., College Station, Texas, USA).

## Results

The electronic databases searches identified 11,565 potential references. After removing duplicates, applying inclusion and exclusion criteria, and excluding irrelevant articles based on the title and abstract, 260 articles were included in the full-text review. Of these, 20 articles with a total sample of 89,928 patients were included for meta-analysis (Fig 1).

### Characteristics of articles included

The characteristics of the 20 articles included in the analysis are presented in Tables 1–3. Among these studies, 13 were conducted in the ED [22–29], including five articles on patients presenting with sepsis alone [30–34], and seven in the prehospital setting [9, 35–40]. The study designs included prospective cohort (n = 11) [23, 25–30, 33, 36–38], retrospective cohort (n = 8) [9, 22, 24, 31, 32, 34, 35, 39], and pragmatic clinical trial design (n = 1) [40]. In the ED setting, ten studies used MEWS [22–28, 30–32], and four studies used NEWS [24, 29, 33, 34]. Five studies evaluated NEWS2 [36–40] in the prehospital setting, and three studies used NEWS [9, 35, 40]. Overall, the meta-analysis included 18,270 and 71,658 patients from the ED and prehospital settings, respectively.

### Risk of bias and quality assessment

Quality and risk of bias assessments of each included study are found in S3 File. A sensitivity analysis was conducted with and without studies with high risk of bias in the ED setting, as shown in Figs 2 and 3. Of all studies, two were rated poor (10%), and the remaining 18 (90%) were rated good.

### Meta-analysis: ED setting

The pooled diagnostic ORs (DOR) of MEWS (cut-off points of $\geq 3$ and $\geq 4$) and NEWS (cut-off point $\geq 6$) to predict 30-day mortality and of MEWS (cut-off point $\geq 3$) to predict ICU admission were estimated (Fig 2). The lowest and highest DORs to predict 30-day mortality was 4.05 (95% confidence interval (CI) 2.35–6.99, $I^2$ = 73.0%) for MEWS with a cut-off point

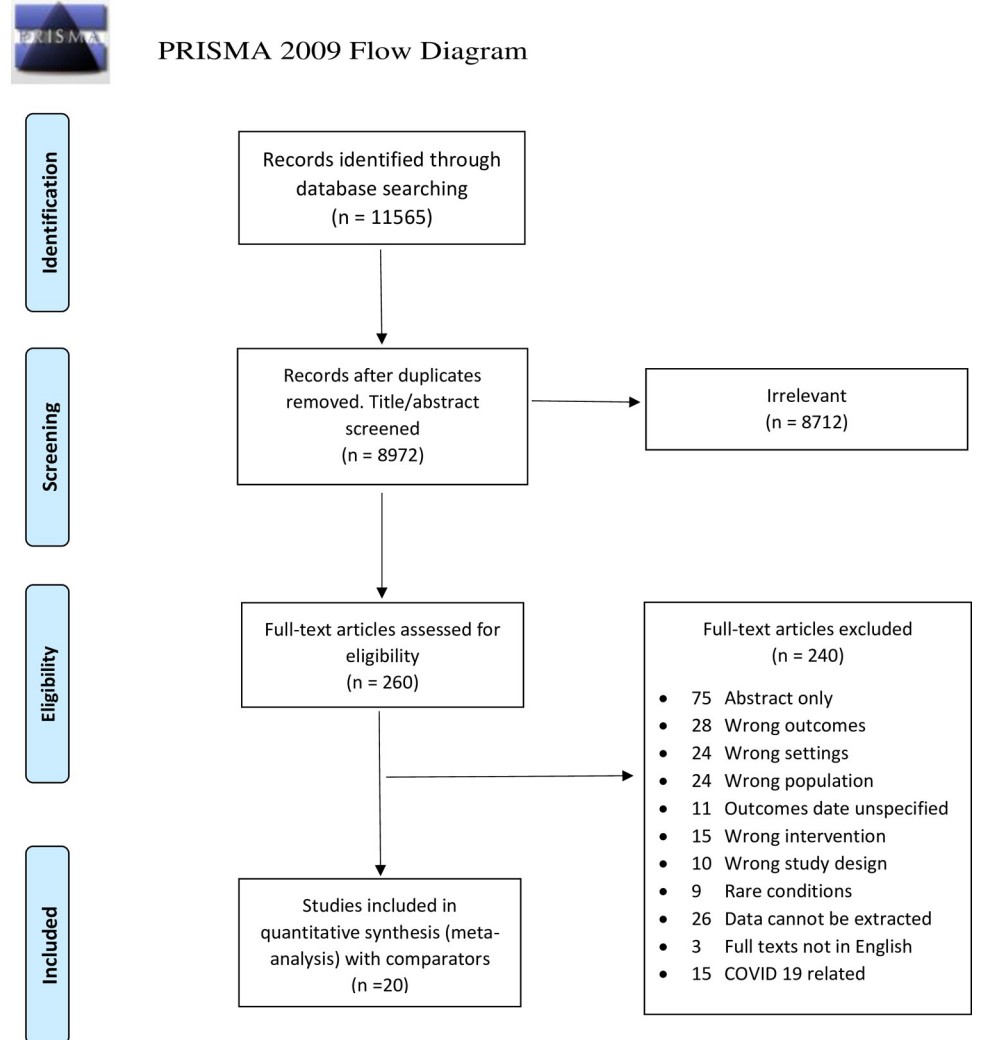

PRISMA 2009 Flow Diagram

**Identification**

Records identified through database searching
(n = 11565)

**Screening**

Records after duplicates removed. Title/abstract screened
(n = 8972)

Irrelevant
(n = 8712)

**Eligibility**

Full-text articles assessed for eligibility
(n = 260)

Full-text articles excluded
(n = 240)

- 75 Abstract only
- 28 Wrong outcomes
- 24 Wrong settings
- 24 Wrong population
- 11 Outcomes date unspecified
- 15 Wrong intervention
- 10 Wrong study design
- 9 Rare conditions
- 26 Data cannot be extracted
- 3 Full texts not in English
- 15 COVID 19 related

**Included**

Studies included in quantitative synthesis (meta-analysis) with comparators
(n =20)

**Fig 1. PRISMA chart.**

of $\geq$ 3 and 6.48 (95% CI 1.83–22.89, $I^2$ = 90%) for MEWS with a cut-off point of $\geq$ 4. A similar DOR for NEWS at a cut-off point $\geq$6 was found [4.92 (95% CI 2.71–8.96, $I^2$ = 65.5%)]. MEWS at a cut-off point of $\geq$ 3 also predicted admission to ICU with a pooled DOR of 5.54 (95% CI 2.02–15.21, $I^2$ = 50.9%). The confidence intervals of the highest and lowest pooled ORs overlapped, and no statistically significant differences were detected between them, p = 0.757. For patients experiencing sepsis only, when assessing 30-days mortality, MEWS with a cut-off $\geq$5 had a DOR of 3.05 (95% CI 2.00–4.65, I2 = 0%), and the DOR was 4.74 (95% CI 4.08–5.50, $I^2$ = 0.0%) for NEWS $\geq$7.

## Meta-analysis: Prehospital setting

As illustrated in Fig 4, with cut-off points at 5, 7, and 9, NEWS2 had DORs of 14.06 (95% CI 9.09–21.75, $I^2$ = 0%,), 12.26 (95%, CI 8.58–17.64, $I^2$ = 4.4%,), and 20.37 (95% CI 13.16–31.52, $I^2$ = 0%), respectively to predict short-term mortality. NEWS at a cut-off point $\geq$7 had a DOR of 11.63 (95%, CI 9.75–13.88, $I^2$ = 0%) to predict this outcome, which was not statistically different than NEWS2 with the same cut-off point. However, increasing the cut-off point to 9

**Table 1. Description of included studies: ED setting.**

| Study | Inclusion criteria | Exclusion criteria | Mean/ Median age (years) | % Male | Sample size | Outcome | EWS | Cut-off points | SEN % | SPE % | AUC/ OR (95% CI) | ROB |
|---|---|---|---|---|---|---|---|---|---|---|---|---|
| Jiang et al. 2019 [22] | Multiple trauma | Age < 16yrs and missing data, DOA, medical patients | 48.0 | 73.7 | 1127 | 28 days in-hospital mortality | MEWS | ≥3 | 93.0 | 75.0 | 0.78 | Poor |
| Demircan et al. 2020 [23] | Age ≥65yrs, Yellow or red triage code | TRI, CPR prior arrival, loss of contacts | 77.2 | 52.0 | 1106 | 28 days in-hospital mortality | MEWS | ≥3 | 58.4 | 65.3 | 0.65 | Good |
| Mitsunaga et al. 2019 [24] | Age ≥65yrs | N/S | 78.0 | 53.9 | 2204 | 28 days in-hospital mortality | NEWS | ≥5 | 78.7 | 64.0 | 0.79 | Good |
| | | | | | | 28 days in-hospital mortality | MEWS | ≥3 | 69.3 | 67.6 | 0.72 | |
| Koksal et al. 2016 [25] | Age ≥18yrs, triage category 1 or 2 | Age <18yrs and TRI | 62.0 | 49.4 | 502 | 28 days in-hospital mortality | MEWS | ≥3 | 78.0 | 79.9 | 0.85 | Poor |
| Maftoohian et al. 2020 [26] | Age ≥18yrs | Advance airway, CPR, intubation applied, not admitted, DOA | 62.1 | 50.9 | 381 | 30 days in-hospital mortality | MEWS | ≥3 | 78.3 | 68.4 | 0.73 | Good |
| | | | | | | 30 days in-hospital mortality | MEWS | ≥4 | 30.4 | 83.2 | N/S | |
| | | | | | | ICU admission | MEWS | ≥3 | 85.7 | 67.6 | 0.77 | |
| Yuan et al. 2018 [27] | Stay longer than 24 hours | Age < 14yrs, LOS less than 24 hours, incomplete information, lost follow up, non-cooperative family members | 64.0 | 60.0 | 612 | 28 days in-hospital mortality | MEWS | ≥4 | 56.9 | 79.4 | 0.72 | Good |
| | | | | | | ICU admission | MEWS | ≥3 | 64.5 | 68.7 | 0.73 | |
| Dundar et al. 2016 [28] | Age ≥65yrs | Age < 65yrs and TRI patients, CPR | 75.0 | 55.9 | 671 | 28 days in-hospital mortality | MEWS | ≥4 | 74.0 | 89.0 | 0.89 | Good |
| Graham et al. 2020 [29] | Age ≥18yrs with triage as Emergency or Urgent | Age <18yrs, pregnant, presenting out of research hours | 72.0 | 50.9 | 1253 | 30 days in-hospital mortality | NEWS | ≥5 | 35.8 | 86.8 | 0.61 | Good |

significantly improved the predictability of the tool. Conversely, in the prehospital setting, NEWS could not accurately predict 30-day mortality [DOR of 2.58 (95% CI 0.59–11.21)].

## Between-study variances

Studies reporting 30-day mortality had moderate to high heterogeneity ($I^2 > 70\%$) in the ED and prehospital settings. Meta-regressions, including studies from the ED setting, were constructed to detect the variables, potentially contributing to the between-studies differences. Only patients' age contributed to the heterogeneity of the study variables, explaining 92% of the between-study variance. We could not perform meta-regression in the prehospital setting due to the limited number of studies investigating 30-day mortality.

## Publication bias

Since studies often reported multiple results for different EWS scores. We included the highest and lowest reported ORs in each study in two separated Deeks' funnel asymmetry tests to

**Table 2. Description of included studies: ED setting, patients presenting with sepsis only.**

| Study | Inclusion criteria | Exclusion criteria | Mean/Median age (years) | % Male | Sample size | Outcome | EWS | Cut-off points | SEN % | SPE % | AUC/OR (95% CI) | ROB |
|---|---|---|---|---|---|---|---|---|---|---|---|---|
| Geier, 2013 [30] | Patient age≥65 with suspect sepsis | Unclear | 68.3 | 54.3 | 151 | 28 days in-hospital mortality | MEWS | ≥5 | 42.9 | 74.4 | 0.642 | Good |
| van der Woude, 2018 [31] | Patient age ≥18 with suspect sepsis | Missing data | 55.3 | 50.3 | 577 | 28 days in-hospital mortality | MEWS | ≥5 | 23.8 | 87.0 | N/S | Good |
| Vorwerk, 2008 [32] | Patient age ≥16 with suspect sepsis | Missing data | 69.7 | 51.0 | 307 | 28 days in-hospital mortality | MEWS | ≥5 | 72.2 | 59.2 | 0.72 | Good |
| Saeed, 2019 [33] | Patient age ≥18 with suspect sepsis | Pregnancy or refusal to participate | 63.3 | 50.4 | 1175 | 28 days in-hospital mortality | NEWS | ≥7 | 59.0 | 74.0 | 0.72 | Good |
| Brink, 2019 [34] | Patient age ≥18 with suspect sepsis | TRI | 57.0 | 55.8 | 8204 | 10 days in-hospital mortality | NEWS | ≥7 | 76.3 | 65.9 | 0.84 | Good |

detect publication bias, as shown in Figs 5 and 6. No evidence for publication bias was detected, with p values using the highest or lowest ORs p = 0.37 and p = 0.69, respectively.

## Discussion

While the application of EWS scores in the ED is well documented, only limited studies assessed their function and applicability in the prehospital setting. This systematic review and meta-analysis investigated the predictability of various EWS scores, as utilised in the ED or prehospital setting, to predict up-to-3- and 30-day mortality and admission to ICU. We found that different cut-off points of various EWS scores applied in the ED had comparable abilities to predict clinical decline, including death among hospitalised patients. Conversely, in the pre-hospital setting with relatively high cut-off points, the EWS only predicted short-term clinical deterioration failing to predict 30-day mortality.

Similar to other studies, our systematic review demonstrated the ability of EWS scores to predict 30-day mortality when applied in the ED setting [41, 42]. However, unlike other studies that suggested optimal cut-off points of different EWS scores to predict clinical deterioration [43–47], this systematic review did not detect any significant variation in the predictability of different scores by different cut-off points when applied in the ED. The choice of cut-off point may depend on the severity of illness and the acuteness of the investigated condition. Patient presentation varies across different settings; in the prehospital setting, the general population attended by paramedics are less severely ill (with a considerable majority having non-urgent low acuity presentations) than patients who were sick enough to have been brought to the ED [48]. In critically ill patients, lower cut-off points are able to predict clinical deterioration [16, 49, 50], which may indicate that the predictability of the score is outcome and patient population specific, as evidenced by this systematic review with both settings requiring different optimal thresholds. The cut-off points applied in ED are typically lower than those used in the prehospital setting. The high cut-off points in the prehospital setting may imply that EWS scores are adequate to predict short-term clinical deterioration among the critically ill, as the higher EWS scores target more severely ill patients who are at high risk of clinical deterioration in the short term [16, 49, 50]. Thus, the high thresholds used in the prehospital setting may be targeting the critically ill patients who will constitute a relatively small proportion of the overall prehospital patient population that paramedics treat. The

**Table 3. Description of included studies: Prehospital setting.**

| Study | Inclusion criteria | Exclusion criteria | Mean/ Median age (years) | % Male | Sample size | Outcome | EWS | Cut-off points | SEN % | SPE % | AUC/ OR (95% CI) | ROB |
|---|---|---|---|---|---|---|---|---|---|---|---|---|
| Pirneskoski et al. 2019 [35] | Age ≥18yrs | Missing data, data errors, without Finnish ID | 65.8 | 47.5 | 35800 | 3 days in-hospital mortality | NEWS | ≥7 | 77.0 | 77.1 | 0.84 | Good |
| | | | | | | 30 days in-hospital mortality | NEWS | ≥7 | 75.9 | 63.4 | 0.76 | |
| Martín-Rodriguez et al. 2019 [36] | Age ≥18yrs, attended by ALSU and transported to ED | Age <18yrs, CPR, DOA pregnancy, psychiatric, palliative, or discharge in situ | 66.0 | 58.5 | 349 | 3 days in-hospital mortality | NEWS2 | ≥9 | 88.0 | 80.0 | N/S | Good |
| Martín-Rodriguez et al. 2020 [37] | Age ≥18yrs, attended by ALSU and transported to ED | Age <18yrs, CPR, DOA pregnancy, psychiatric, palliative, or discharge in situ, loss of follow-ups | 69.0 | 58.9 | 2335 | 3 days in-hospital mortality | NEWS2 | ≥9 | 74.8 | 85.2 | 0.86 | Good |
| Martín-Rodriguez et al. 2019 [38] | Age ≥18yrs, attended by ALSU and transported to ED | Age <18yrs, CPR, DOA pregnancy, psychiatric, palliative, or discharge in situ | 68.0 | 59.5 | 1288 | 3 days in-hospital mortality | NEWS2 | ≥9 | 79.7 | 84.5 | 0.87 | Good |
| | | | | | 1026 | 3 days in-hospital mortality-medical only | NEWS2 | ≥7 | 87.5 | 71.8 | N/S | |
| Vihonen et al. 2020 [9] | Age ≥18yrs | Age <18yrs | 69.0 | 48.0 | 27141 | 3 days in-hospital mortality | NEWS | ≥7 | 71.0 | 83.0 | 0.84 | Good |
| | | | | | | 30 days in-hospital mortality | NEWS | ≥7 | 51.0 | 54.0 | 0.75 | |
| Magnusson et al. 2020 [39] | Age ≥16yrs, attended by ALSU and transported to ED | Missing data, DOA | 69.0 | 48.0 | 473 | 48 hours in-hospital mortality | NEWS2 | ≥5 | 72.7 | 81.9 | 0.77 | Good |
| Martín-Rodríguez et al.2020 [40] | Age ≥18yrs, attended by ALSU and transported to ED | Cardiac arrest, terminally ill, pregnant, or not transported by ALSU | 69.0 | 58.9 | 3273 | 3 days in-hospital mortality | NEWS2 | ≥7 | 78.1 | 75.9 | 11.2 (7.5–16.7) | Good |
| | | | | | | 3 days in-hospital mortality | NEWS2 | ≥5 | 91.2 | 60.0 | 15.5 (8.9–26.9) | |
| | | | | | | 3 days in-hospital mortality | NEWS | ≥7 | 78.9 | 76.6 | 12.3 (7.7–19.4) | |

ALSU, advanced life support units; BLS, basic life support; CPR, cardio-pulmonology resuscitation commenced; CFS, clinical frailty scale; DOA, dead on arrival; HRV, heart rate variability; ENT, ears, nose, throat related patients; IHCA, in-hospital cardiac arrest; LOC, loss of consciousness; LOS, length of stay; OHCA, out of hospital cardiac arrest; PACS, Patient Acuity Category Scale; TRI, trauma-related injuries; NFR, not for resuscitation. AUC, Area under the ROC Curve; OR, Odds ratios; Sen%, sensitivity, Spe%, specificity, ROB, risk of bias. N/S, Not stated. NEWS, National Early Warning Score; NEWS2, National Early Warning Score 2; MEWS, Modified Early Warning Score.

review conducted by Williams et al. on the use of EWS scores in the prehospital setting also suggests that critically ill patients are the best candidates for the use of prehospital EWS scores [16]. The authors also argue that achieving an optimal EWS score is difficult due to the short duration of the interaction between paramedics and patients. The reporting of high cut-off points in the prehospital setting is also due to a trade-off in sensitivity and specificity. Lower

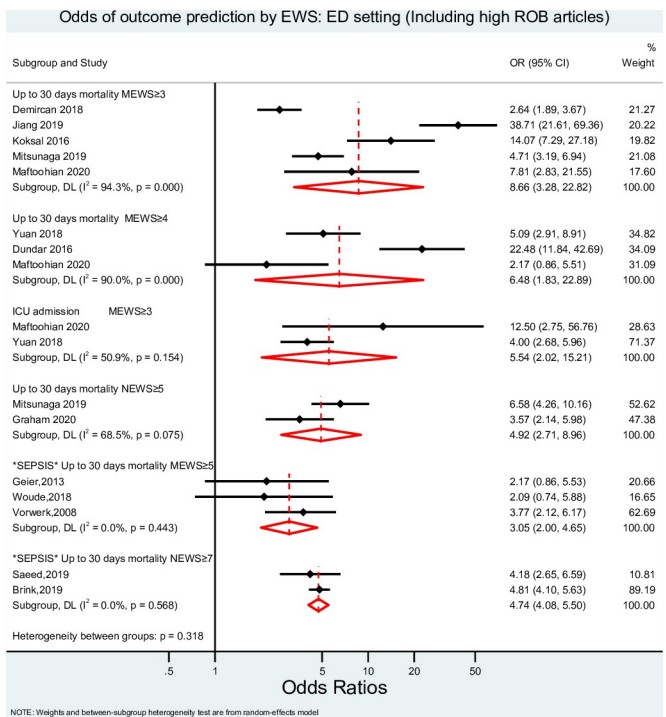

**Fig 2. Meta-analysis result for ED setting (Including studies with high ROB).** EWS, Early Warning System score; OR, odds ratio; MEWS, Modified Early Warning Score; NEWS, National Early Warning Score; ICU, Intensive Care Unit; ROB, Risk of Bias.

cut-off points often result in poor sensitivity and specificity in the prehospital setting. Conversely, the cut-off points in the ED are often similar to the cut-off points used in in-hospital settings suggesting that EWS scores can be compared between the ED and in-hospital wards, whereas this comparison becomes less valid when it is conducted against the prehospital setting. In our review, NEWS2 with a cut-off point of 5, 7 or 9 and NEWS with a cut-off point of 7 had similar predictability. This is supported by a study that compared NEWS and NEWS2 at the same threshold of 7 without detecting significant difference between the two scores when predicting short-term mortality [3]. The Royal College of Physicians in London argue that NEWS2 is superior to NEWS in predicting clinical deterioration [51, 52]; however, this was not supported by our findings. Similarly, Hodgson and colleagues demonstrated that NEWS2 did not outperform NEWS in predicting clinical deterioration in patients admitted to hospital with acute exacerbation of chronic obstructive pulmonary disease, [52], which was one of the main reasons why chronic hypoxia patient presentations were as an additional parameter in NEWS2. Based on the available studies and our systematic review, NEWS and NEWS2 had similar predictabilities.

In this review, the analysis on patients experiencing sepsis was conducted separately partly because the diagnosis of sepsis requires pre-defined cut-off values for the vital signs. Patients experiencing sepsis may differ from other patients in such that they will have higher baseline EWS scores. For example, common criteria for sepsis include presence of two or more of the following: temperature > 38˚C or < 36˚C, heart rate > 90/min, respiratory rate > 20/min or PaCO2 < 32 mm Hg (4.3 kPa), white blood cell count > 12 000/mm$^3$ or <4000/mm$^3$ or > 10% cells with immature bands [53]. These criteria indicate systemic clinical decline and may result in elevated EWS scores. Our systematic review shows that the predictive ability of

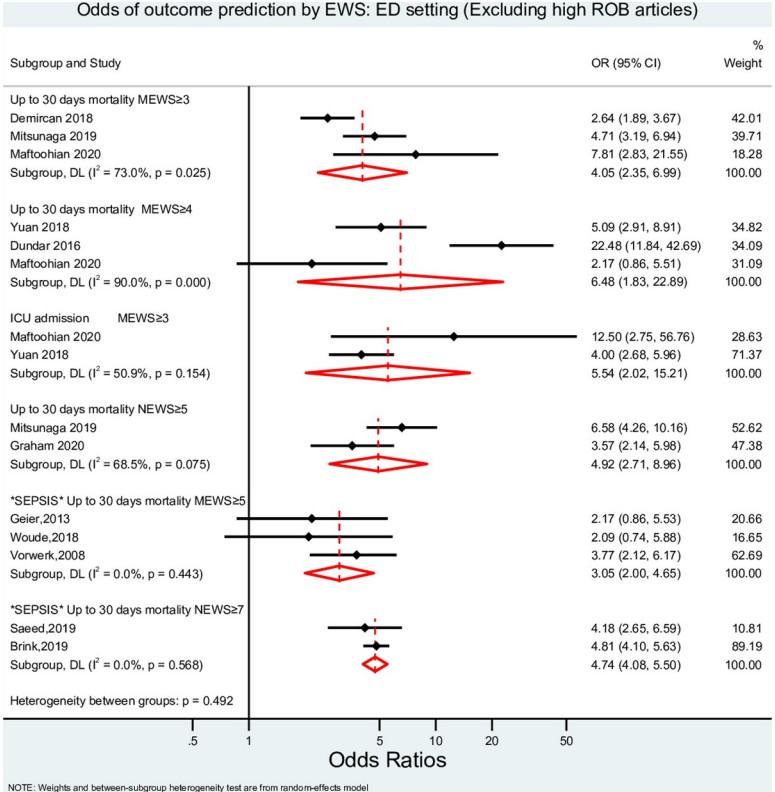

**Fig 3. Meta-analysis result for ED setting (Excluding studies with high ROB).** EWS, Early Warning System score; OR, odds ratio; MEWS, Modified Early Warning Score; NEWS, National Early Warning Score; ICU, Intensive Care Unit; ROB, Risk of Bias.

EWS scores of long-term mortality for patients experiencing sepsis is not different from the mainstream ED patient population.

## Limitations

The results of the systematic review apply only to the EWS scores assessed in this study. The analysis lacked the power to evaluate medical *versus* trauma conditions separately. Patients' main complaints and diagnoses were unknown and could not be accounted for. The number of studies reporting cardiac and/or respiratory arrest and length of stay were limited and could not be included for meta-analysis. We also acknowledge the lack of reporting of short-term mortality in ED. The articles that reported short-term mortality failed to include cut-off points, making the authors unable to draw conclusions based on available data. Confounding factors that may have affected long-term mortality following hospital admission were unknown and were not accounted for.

## Conclusions

The accuracy and predictability of the EWS scores depend on several factors, such as the outcome measure, population, and the types of clinical settings. In the ED setting, the patient population is by default more morbid than that managed by paramedics in the community. This, in turn, may explain the ability of low EWS cut-off points to predict clinical deterioration in the ED. We report that different EWS cut-off points in the ED have similar predictability.

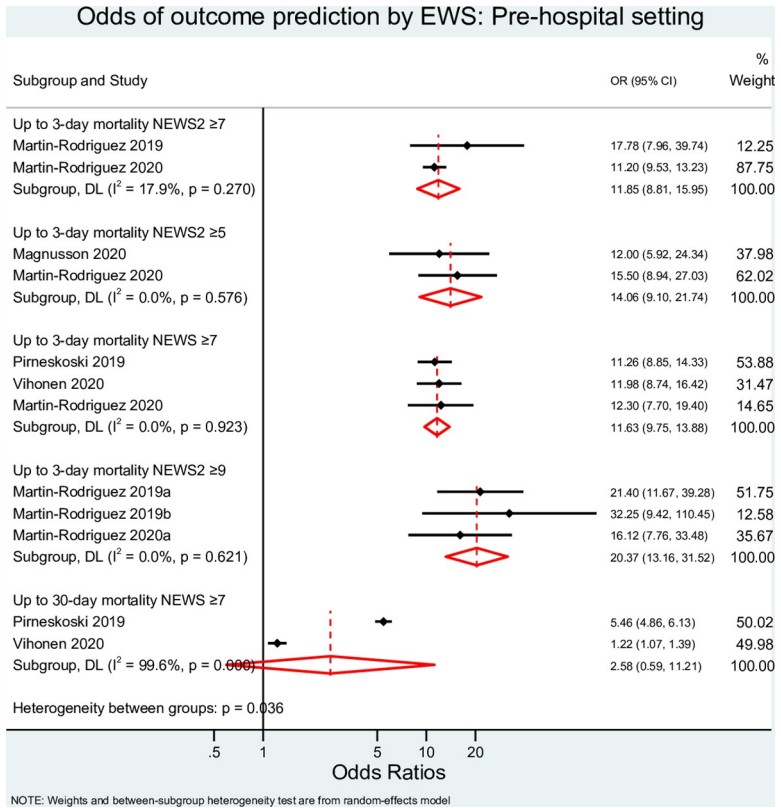

**Fig 4. Meta-analysis result for prehospital setting.** EWS, Early Warning System score; OR, odds ratio; MEWS, Modified Early Warning Score; NEWS, National Early Warning Score; ICU, Intensive Care Unit; ROB, Risk of Bias.

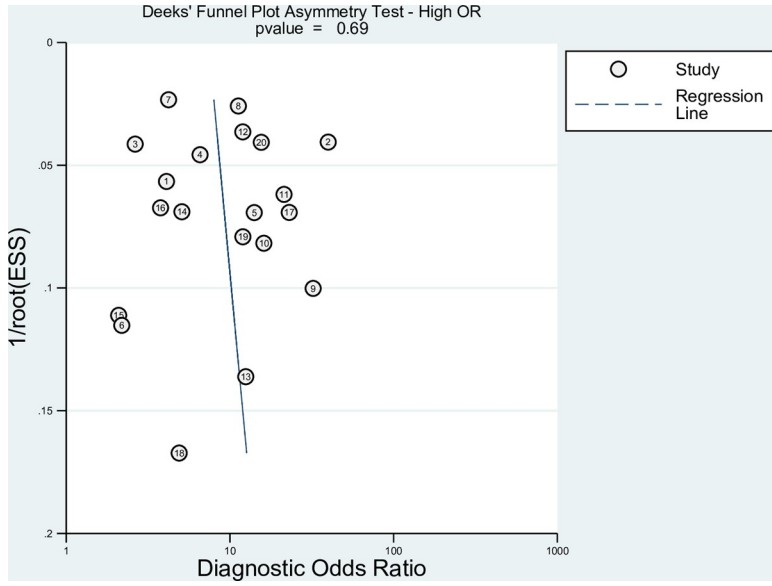

**Fig 5. Deeks' funnel plot testing for publication bias using the highest OR.**

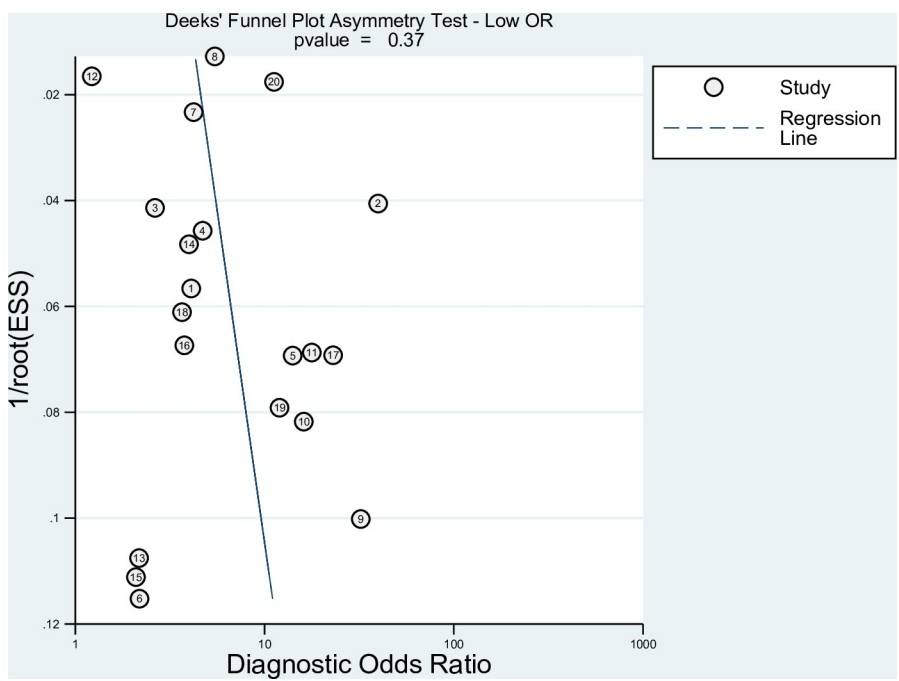

**Fig 6. Deeks' funnel plot testing for publication bias using the lowest OR.**

Studies using EWS scores in the prehospital setting utilised relatively high cut-off points. This may indicate that early warning scoring systems may be less applicable for the general population treated by medical care personnel in the prehospital setting. This scoring system may only be suited for critically ill patients treated in the prehospital setting. Our findings suggest that EWS scores applied in the prehospital setting cannot accurately predict long-term events, including 30-day mortality. However, this SR demonstrates that EWS scores used in the prehospital setting can predict short term clinical decline.

## Supporting information

**S1 Checklist. PRISMA 2020 checklist.**
(DOCX)

**S1 File. A PICO framework was used to inform the literature search strategy.**
(PDF)

**S2 File. Detailed search history.**
(PDF)

**S3 File. Risk of bias.**
(PDF)

## Author Contributions

**Data curation:** Gigi Guan.

**Formal analysis:** Gigi Guan.

**Investigation:** Gigi Guan, Crystal Man Ying Lee.

**Methodology:** Crystal Man Ying Lee, George Mnatzaganian.

**Resources:** Gigi Guan, George Mnatzaganian.

**Supervision:** Crystal Man Ying Lee, George Mnatzaganian.

**Writing – original draft:** Gigi Guan, Crystal Man Ying Lee, George Mnatzaganian.

**Writing – review & editing:** Gigi Guan, Crystal Man Ying Lee, Stephen Begg, Angela Crombie, George Mnatzaganian.

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
