## [Decision Letter · Decision Letter 0]

20 Jan 2022

PONE-D-21-35942The Use of Early Warning System Scores in Prehospital and Emergency Department Settings to Predict Clinical Deterioration: A Systematic Review and Meta-AnalysisPLOS ONE

Dear Dr. Guan,

Thank you for submitting your manuscript to PLOS ONE. After careful consideration, we feel that it has merit but does not fully meet PLOS ONE’s publication criteria as it currently stands. Therefore, we invite you to submit a revised version of the manuscript that addresses the points raised during the review process.

We look forward to receiving your revised manuscript.

Kind regards,

Yong-Hong Kuo

Academic Editor

PLOS ONE

Journal Requirements:

2. We noted in your submission details that a portion of your manuscript may have been presented or published elsewhere. (This manuscript was previously submitted to a different PLOS journal as either a presubmission inquiry or a full submission.

PLOS Medicine: PMEDICINE-D-21-04630) Please clarify whether this [conference proceeding or publication] was peer-reviewed and formally published. If this work was previously peer-reviewed and published, in the cover letter please provide the reason that this work does not constitute dual publication and should be included in the current manuscript.

3. We note you have included a table to which you do not refer in the text of your manuscript. Please ensure that you refer to Table 1a, 1b, 1c in your text; if accepted, production will need this reference to link the reader to the Table.

Additional Editor Comments:

The manuscript has been reviewed by two experts in the area. Both of them have favorable recommendations and comments. Based on their review reports, I recommend Minor Revision.

Reviewers' comments:

Reviewer's Responses to Questions

**Comments to the Author**

1. Is the manuscript technically sound, and do the data support the conclusions?

Reviewer #1: Yes

Reviewer #2: Yes

2. Has the statistical analysis been performed appropriately and rigorously? 

Reviewer #1: Yes

Reviewer #2: I Don't Know

3. Have the authors made all data underlying the findings in their manuscript fully available?

Reviewer #1: Yes

Reviewer #2: Yes

4. Is the manuscript presented in an intelligible fashion and written in standard English?

Reviewer #1: Yes

Reviewer #2: Yes

5. Review Comments to the Author

Reviewer #1: This is the well compared study showing differences in hospital setting and pre hospital setting. We also use qSOFA for pre hospital and SOFA for emergency.As the study has used sepsis case and we are using SOFA or qSOFA to predict the outcome.Using this predictors also in the early warning score would have made more clear for the reader. As pre hospital setting has been used for patient treated by medical care personnel in this study .Cases from those category also referred when they become critical.And we usually understand the management in EMS while transporting patient to the definitive care center.So it would be good for reader if this has been clear in the initial part about pre hospital.

Reviewer #2: The authors have in their revised version adequately addressed the strengths and limitations in the submitted paper. Although the application of EWS in prehospital settings has many drawbacks and is problematic the authors have clearly stated the choice of thresholds, cut-off-points in different settings: The statement in the conclusion that EWS scores applied in the prehospital setting cannot accurately predict long-term events but can predict short term clinical decline puts the results of the systematic review/metaanalyses in the right perspective.

6. PLOS authors have the option to publish the peer review history of their article (what does this mean?). If published, this will include your full peer review and any attached files.

Reviewer #1: No

Reviewer #2: No

---

## [Author Response · Author response to Decision Letter 0]

9 Feb 2022

Dear Yong-Hong Kuo,

All requests have been addressed in the documents.

---

## [Decision Letter · Decision Letter 1]

4 Mar 2022

The Use of Early Warning System Scores in Prehospital and Emergency Department Settings to Predict Clinical Deterioration: A Systematic Review and Meta-Analysis

PONE-D-21-35942R1

Dear Dr. Guan,

We’re pleased to inform you that your manuscript has been judged scientifically suitable for publication and will be formally accepted for publication once it meets all outstanding technical requirements.

Kind regards,

Yong-Hong Kuo

Academic Editor

PLOS ONE

Additional Editor Comments (optional):

The referees from the previous round have reviewed this revision. Both of them are satisfied with the revision and recommend Accept. Based on their comments and recommendations, I recommend Accept.

Reviewers' comments:

Reviewer's Responses to Questions

**Comments to the Author**

1. If the authors have adequately addressed your comments raised in a previous round of review and you feel that this manuscript is now acceptable for publication, you may indicate that here to bypass the “Comments to the Author” section, enter your conflict of interest statement in the “Confidential to Editor” section, and submit your "Accept" recommendation.

Reviewer #1: All comments have been addressed

Reviewer #2: All comments have been addressed

2. Is the manuscript technically sound, and do the data support the conclusions?

Reviewer #1: Yes

Reviewer #2: (No Response)

3. Has the statistical analysis been performed appropriately and rigorously? 

Reviewer #1: Yes

Reviewer #2: (No Response)

4. Have the authors made all data underlying the findings in their manuscript fully available?

Reviewer #1: Yes

Reviewer #2: (No Response)

5. Is the manuscript presented in an intelligible fashion and written in standard English?

Reviewer #1: Yes

Reviewer #2: (No Response)

6. Review Comments to the Author

Reviewer #1: As an emergency physician i found it very useful in the ED.MEWS, and NEWS at cut-off points of 3, 4, or 6 had similar pooled diagnostic odds to predict 30-day mortality from this study. Further study with SOFA and APACHE II score will helps in ED in coming days.As predicting outcome is sepsis with NEWS and MEWS it also helps correlates with SOFA as diagnosis of sepsis according to Surviving Sepsis Campaign guideline from SOFA score.

Reviewer #2: (No Response)

7. PLOS authors have the option to publish the peer review history of their article (what does this mean?). If published, this will include your full peer review and any attached files.

Reviewer #1: **Yes: **Laxman Bhusal

Reviewer #2: No

---

## [Editor Report · Acceptance letter]

8 Mar 2022

PONE-D-21-35942R1 

The use of early warning system scores in prehospital and emergency department settings to predict clinical deterioration: a systematic review and meta-analysis 

Dear Dr. Guan:

I'm pleased to inform you that your manuscript has been deemed suitable for publication in PLOS ONE. Congratulations! Your manuscript is now with our production department. 

Kind regards, 

on behalf of

Dr. Yong-Hong Kuo 

Academic Editor

PLOS ONE